# Deep Multi-State Dynamic Recurrent Neural Networks Operating on Wavelet Based Neural Features for Robust Brain Machine Interfaces

**Benyamin Haghi**[1,*], **Spencer Kellis**[2], **Sahil Shah**[1], **Maitreyi Ashok**[1], **Luke Bashford**[2], **Daniel Kramer**[3], **Brian Lee**[3], **Charles Liu**[3], **Richard A. Andersen**[2], **Azita Emami**[1]

1 Electrical Engineering Department, Caltech, Pasadena, CA, USA
2 Biology and Biological Engineering Department, Caltech, Pasadena, CA, USA
3 Neurorestoration Center and Neurosurgery, USC Keck School of Medicine, L.A., CA, USA

*benyamin.a.haghi@caltech.edu

## Abstract

We present a new deep multi-state Dynamic Recurrent Neural Network (DRNN) architecture for Brain Machine Interface (BMI) applications. Our DRNN is used to predict Cartesian representation of a computer cursor movement kinematics from open-loop neural data recorded from the posterior parietal cortex (PPC) of a human subject in a BMI system. We design the algorithm to achieve a reasonable trade-off between performance and robustness, and we constrain memory usage in favor of future hardware implementation. We feed the predictions of the network back to the input to improve prediction performance and robustness. We apply a scheduled sampling approach to the model in order to solve a statistical distribution mismatch between the ground truth and predictions. Additionally, we configure a small DRNN to operate with a short history of input, reducing the required buffering of input data and number of memory accesses. This configuration lowers the expected power consumption in a neural network accelerator. Operating on wavelet-based neural features, we show that the average performance of DRNN surpasses other state-of-the-art methods in the literature on both single- and multi-day data recorded over 43 days. Results show that multi-state DRNN has the potential to model the nonlinear relationships between the neural data and kinematics for robust BMIs.

## 1 Introduction

Brain-machine interfaces (BMIs) can help spinal cord injury (SCI) patients by decoding neural activity into useful control signals for guiding robotic limbs, computer cursors, or other assistive devices [1]. BMI in its most basic form maps neural signals into movement control signals and then closes the loop to enable direct neural control of movements. Such systems have shown promise in helping SCI patients. However, improving performance and robustness of these systems remains challenging. Even for simple movements, such as moving a computer cursor to a target on a computer screen, decoding performance can be highly variable over time. Furthermore, most BMI systems currently run on high-power computer systems. Clinical translation of these systems will require decoders that can adapt to changing neural conditions and which operate efficiently enough to run on mobile, even implantable, platforms.

Conventionally, linear decoders have been used to find the relationship between kinematics and neural signals of the motor cortex. For instance, Wu et al. [2] use a linear model to decode the neural activity of two macaque monkeys. Orsborn et al. [3] apply a Kalman filter, updating the model on batches of neural data of an adult monkey, to predict kinematics in a center-out task. Gilja et al. [4] propose a Kalman Filter to predict hand movement velocities of a monkey in a center-out task. However, all of these algorithms can only predict piecewise linear relationships between the neural data and kinematics. Moreover, because of nonstationarity and low signal-to-noise ratio (SNR) in the neural data, linear decoders need to be regularly re-calibrated [2].

Recently, nonlinear machine learning algorithms have shown promise in attaining high performance and robustness in BMIs. For instance, Wessberg et al. [5] apply a fully-connected neural network to neural data recorded from a monkey. Shpigelman et al. [6] show that a Gaussian kernel outperforms a linear kernel in a Kernel Auto-Regressive Moving Average (KARMA) algorithm when decoding 3D kinematics from macaque neural activity. Sussillo et al. [7] apply a large FORCE Dynamic Recurrent Neural Network (F-DRNN) on neural data recorded from the primary motor cortex in two monkeys, and then they test the stability of the model over multiple days [8]. Zhang et al. [9] and Schwemmer et al. [10] extract wavelet based features of motor cortex neural data of a human subject to classify intended hand movements by using a nonlinear support vector machine (SVM) and a large deep neural network, respectively. Hosman et al. [11] pass motor cortex neural firing rates to an LSTM and a Kalman filter to compare their performances for decoding intended cursor velocity of a human subject. These nonlinear learning-based decoders have shown more stability over multiple days and have improved performance compared to prior linear methods. However, they all have been applied to motor cortex data by mostly using neural firing rates as input features, which show more variability over long periods [2]. Recent work has demonstrated that neural activity in the posterior parietal cortex (PPC) can be used to support BMIs [12, 13, 14, 15, 16, 17, 18], although the encoding of movement kinematics appears to be complex. PPC processes a rich set of high-level aspects of movement including sensory integration, planning, and execution [13] and may encode this information differently [15]. These characteristics of PPC differentiate it from other brain areas and, while providing a large amount of information to the decoder, also require new paradigms, such as those discussed here, to extract useful information. Therefore, extracting appropriate neural features and designing a robust decoder that can model this relationship in an actual BMI setting is required.

We propose a new Deep Multi-State Dynamic Recurrent Neural Network (DRNN) decoder to address the challenges of performance, robustness, and potential hardware implementation. We refer to two theorems to show the stability, convergence, and potential of DRNNs for approximation of state-space trajectories (see supplementary material). We train the DRNN by passing a history of input data to it and feeding the predictions of the system back to the input to improve performance and robustness for sequential data prediction. Moreover, we apply scheduled sampling to solve the statistical distribution discrepancy between the ground truth and predictions. By extracting different neural features, we compare the performance and robustness of the DRNN with the existing methods in the literature to predict hand movement kinematics from open-loop neural data. Our BMI data are recorded from the PPC of a human subject over 43 days. Finally, we discuss the potential for implementing our DRNN efficiently in hardware for implantable platforms. To the best of our knowledge, this is the first demonstration of applying learning-based decoders to a human PPC activity. Our results indicate that the Deep Multi-State DRNN operating on mid-band wavelet-based neural features has the potential to model the nonlinear relationships between the neural data and kinematics for robust BMIs.

## 2 Deep multi-state dynamic recurrent neural network

A DRNN is a nonlinear dynamic system described by a set of differential or difference equations. It contains both feed-forward and feedback synaptic connections. In addition to the recurrent architecture, a nonlinear and dynamic structure enables it to capture time-varying spatiotemporal relationships in the sequential data. Moreover, because of state feedback, a small recurrent network can be equivalent to a large feed-forward network. Therefore, a recurrent network will be computationally efficient, especially for the applications that require hardware implementation [19]. We define our deep multi-state DRNN at each time step $k$ as below:

$$
\begin{cases}
s_k = W_{ss}s_{k-1} + W_{sr}r_{k-1} + W_{si}u_k + W_{sf}z_{k-1} + b_s \\
r_k = tanh(s_k) \\
h_k^{(1)} = tanh(W_{h^{(1)}h^{(1)}}h_{k-1}^{(1)} + W_{h^{(1)}r}r_k + b_{h^{(1)}}) \\
h_k^{(i)} = tanh(W_{h^{(i)}h^{(i)}}h_{k-1}^{(i)} + W_{h^{(i)}h^{(i-1)}}h_k^{(i-1)} + b_{h^{(i)}}) \\
\hat{y}_k = W_{yh^{(l)}}h_k^{(l)} + b_y \\
\hat{y}_k = tanh(\hat{y}_k), \ |\hat{y}_k| > 1 \\
z_k \leftarrow \hat{y}_k \ or \ y_k \ \text{(Scheduled Sampling during Training)}
\end{cases}
\tag{1}
$$

$s \in \mathbb{R}^N$ is the activation variable, and $r \in \mathbb{R}^N$ is the vector of corresponding firing rates. These two internal states track the first- and zero-order differential features of the system, respectively. Unlike

conventional DRNNs, $W_{ss} \in \mathbb{R}^{N \times N}$ generalizes the dynamic structure of our DRNN by letting the network learn the matrix relationship between present and past values of $s$. $W_{sr} \in \mathbb{R}^{N \times N}$ describes the relationship between $s$ and $r$. $W_{su} \in \mathbb{R}^{N \times I}$ relates $s$ to the input vector $u$. $z \in \mathbb{R}^M$ models the added prediction feedback in our DRNN. $W_{sf} \in \mathbb{R}^{N \times M}$ tracks the effect of $z$ on $s$. $i \in \{2, ..., l\}$ and $l$ is the number of layers, $N_i$ is the number of hidden units in $i^{th}$ layer, $h^{(i)} \in \mathbb{R}^{N_i}$ is the hidden state of the $i^{th}$ hidden layer, $W_{h^{(1)}r} \in \mathbb{R}^{N_1 \times N}$, $W_{h^{(i)}h^{(i)}} \in \mathbb{R}^{N_i \times N_i}$, $W_{h^{(i)}h^{(i-1)}} \in \mathbb{R}^{N_i \times N_{i-1}}$, $W_{yh^{(l)}} \in \mathbb{R}^{M \times N_l}$, $b_s \in \mathbb{R}^N$, $b_{h^{(i)}} \in \mathbb{R}^{N_i}$ are the weights and biases of the network. All the parameters are learnable in our DRNN. Although feed-forward neural networks usually require a deep structure, DRNNs generally need fewer than three layers. Algorithm 1 shows the training procedure[1]. Inference is performed by using equation 1. Figure 1 shows the schematic of a two layer DRNN operating on a sample sequence of input data with length $\Delta k$.

During inference, since the ground truth values are unavailable, the feedback, $z_k$, has to be replaced by the previous network predictions. However, the same approach cannot be applied during training since the DRNN has not been trained yet and it may cause poor performance of the DRNN. On the other hand, statistical discrepancies between ground truth and predictions mean that prior ground truth cannot be passed to the input. Because of this disparity between training and testing, the DRNN may enter unseen regions of the state-space, leading to mistakes at the beginning of the sequence prediction process. Therefore, we should find a strategy to start from the ground truth distribution and move toward the predictions' distribution slowly as the DRNN learns.

There exist several approaches to address this issue. Beam search generates several target sequences from the ground truth distribution [20]. However, for continuous state-space models like recurrent networks, the effective number of generated sequences remains small. SEARN is a batch approach that trains a new model according to the current policy at each iteration. Then, it applies the new model on the test set to generate a new policy which is a combination of the previous policy and the actual system behavior [21]. In our implementation, we apply scheduled sampling which can be implemented easily in the online case and has shown better performance than others [22].

In scheduled sampling, at the $i^{th}$ epoch of training, the model pseudorandomly decides whether to feed ground truth (probability $p_i$) or a sample from the predictions' distribution (probability $(1 - p_i)$) back to the network, with probability distribution modeled by $P(y_{k-1}|r_{k-1})$. When $p_i = 1$, the algorithm selects the ground truth, and when $p_i = 0$, it works in Always-Sampling mode. Since the model is not well trained at the beginning of the training process, we adjust these probabilities during training to allow the model to learn the predictions' distribution. Among the various scheduling options for $p_i$ [22], we select linear decay, in which $p_i$ is ramped down linearly from $p_s$ to $p_f$ at each epoch $e$ for the total number of epochs, $E$:

$$p_i = \frac{p_f - p_s}{E} e + p_s \qquad (2)$$

## 3 Pre-processing and feature engineering

We evaluate the performance of our DRNN on 12 neural features: High-frequency, Mid-frequency, and Low-frequency Wavelet features (HWT, MWT, LWT); High-frequency, Mid-frequency, and Low-frequency Fourier powers (HFT, MFT, LFT); Latent Factor Analysis via Dynamical Systems (LFADS) features [23]; High-Pass and Low-Pass Filtered (HPF, LPF) data; Threshold Crossings (TCs); Multi-Unit Activity (MUA); and combined MWT and TCs (MWT + TCs) (Table 1).

To extract wavelet features, we use 'db4' mother wavelet on 50ms moving windows of the voltage time series recorded from each channel. Then, the mean of absolute-valued coefficients for each scale is calculated to generate 11 time series for each channel. HWT is formed from the wavelet scales 1 and 2 (effective frequency range $\geq$ 3.75KHz). MWT is made from the wavelet scales 3 to 6 (234Hz - 3.75KHz). Finally, LWT shows the activity of scales 7 to 11 as the low frequency scales ($\leq$ 234Hz).

Fourier-based features are extracted by computing the Fourier transform with the sampling frequency of 30KHz on one-second moving windows for each channel. Then, the band-powers at the same 11 scales of the wavelet features are divided by the total power at the frequency band of 0Hz - 15KHz.

To generate TCs, we threshold bandpass-filtered (250Hz - 5KHz) neural data at -4 times the root-mean

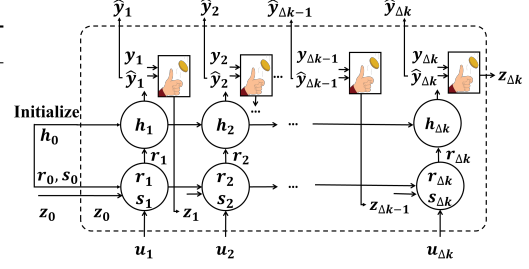

**Algorithm 1** Training – DRNN with Feedback

1: Require: $E, p_f, p_s$
2: **for** $e = 1$ to $E$ **do**
3:     $p_i = \frac{p_f - p_s}{E} e + p_s$
4:     **for** $i = 1$ to $number\ of\ batches$ **do**
5:         Require: $u, y$: Input and ground truth
6:         **if** i = 1 **then**
7:             $z = y$
8:         **end if**
9:         $s \leftarrow \mathcal{N}(0, \sigma_s), r \leftarrow tanh(s)$
10:        **if** $number\ of\ layers = 2$ **then**
11:            $h \leftarrow 0$
12:        **end if**
13:        **for** $k = 2$ to $batch\ length$ **do**
14:            $s_k = W_{ss}s_{k-1} + W_{sr}r_{k-1}$
15:                $+W_{si}u_k + W_{sf}z_{k-1} + b_s$
16:            $r_k = tanh(s_k)$
17:            **if** $layers = 1$ **then**
18:                $\hat{y}_k = W_{yr}r_k + b_y$
19:            **else if** $layers = 2$ **then**
20:                $h_k = tanh(W_{hh}h_{k-1}+W_{hr}r_k+b_h)$
21:                $\hat{y}_k = W_{yh}h_k + b_y$
22:            **end if**
23:            **if** $|\hat{y}_k| > 1$ **then**
24:                $\hat{y}_k = tanh(\hat{y}_k)$
25:            **end if**
26:            $z_k \leftarrow \hat{y}_k$ or $y_k$ (Scheduled Sampling)
27:        **end for**
28:        Update weights and biases: BPTT
29:     **end for**
30: **end for**
31: Until validation loss increases

Figure 1: Training DRNN on a sample sequence of input data with length $\Delta k$.

Table 1: Frequency range of features

| Features | Frequency Range |
|---|---|
| HWT, HFT, HPF | > 3.75KHz |
| TCs, LFADS | 250Hz - 5KHz |
| MWT, MFT, BPF | 234Hz - 3.75KHz |
| LWT, LFT, LPF | < 234Hz |

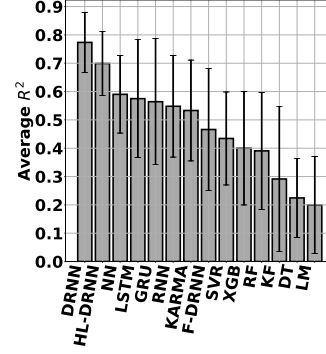

Figure 2: Average performance of decoders operating on MWT over single-day data

-square (RMS) of the noise in each channel. We do not sort the action potential waveforms [24]. Threshold crossing events were then binned at 50ms intervals.

LFADS is a generalization of variational auto-encoders that can be used to model time-varying aspect of neural signals. Pandarinath et al. [23] shows that decoding performance improves when using LFADS to infer smoothed and denoised firing rates. We use LFADS to generate LFADS features based on the trial-by-trial threshold crossings from each center-out task.

To extract HPF, MUA, and LPF features, we apply high-pass, band-pass, and low-pass filters to the broadband data, respectively, by using second-order Chebyshev filters with cut-off frequencies of 234Hz and 3.75KHz. To infer MUA features, we calculate RMS of band-pass filter output. Then, we average the output signals to generate one feature per 50ms for each channel. Table 1 shows the frequency range of features.

We smooth all features with a 1s minjerk smoothing kernel. Afterwards, the kinematics and the features are centered and normalized by the mean and standard deviation of the training data. Then, to select the most informative features for regression, we use XGBoost, which provides a score that indicates how useful each feature is in the construction of its boosted decision trees [25, 26]. In our single-day analysis, we perform Principal Component Analysis (PCA) [27]. Figure 3 shows the block diagram of our BMI system.

## 4 Experimental Results

We conduct our FDA- and IRB-approved study of a BMI with a 32 year-old tetraplegic (C5-C6)

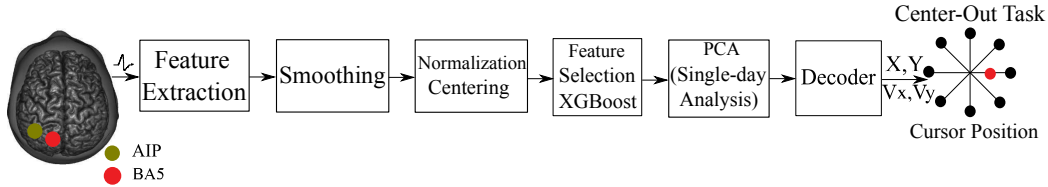

Figure 3: Architecture of our BMI system. Recorded neural activities of Anterior Intraparietal Sulcus (AIP), and Broadman's Area 5 (BA5) are passed to a feature extractor. After pre-processing and feature selection, the data is passed to the decoder to predict the kinematics in a center-out task.

human research participant. This participant has Utah electrode arrays (NeuroPort, Blackrock Microsystems, Salt Lake City, UT, USA) implanted in the medial bank of Anterior Intraparietal Sulcus (AIP), and Broadman's Area 5 (BA5). In a center-out task, a cursor moves, in two dimensions on a computer screen, from the center of a computer screen outward to one of eight target points located around a unit circle. A trial is one trajectory of the cursor from the center of the screen to one of the eight targets on a unit circle (Figure 3). During open-loop training, the participant observes the cursor move under computer control for 3 minutes. We collected open-loop training data from 66 blocks over 43 days for offline analysis of the DRNN. Broadband data were sampled at 30,000 samples/sec from the two implanted electrode arrays (96 channels each). Of the 43 total days, 42 contain 1 to 2 blocks of training data and 1 day contains 6 blocks, with about 50 trials per block. Moreover, these 43 days include 32, 5, 1, and 5 days of 2015, 2016, 2017, and 2018, respectively.

Since the predictions and the ground truth should be close in both micro and macro scales, we report root mean square error (RMSE) and $R^2$ as measures of average point-wise error and the strength of the linear association between the predicted and the ground truth signals, respectively. Results reported in the body of this manuscript are $R^2$ values for Y-axis position. $R^2$ values for X-axis position and velocities in X and Y directions and RMSE values for all the kinematics are all presented in supplementary material. All the curves and bar plots are shown by using $95\%$ confidence intervals and standard deviations, respectively.

The available data is split into train and validation sets for parameter tuning. Parameters are computed on the training data and applied to the validation data. We perform 10 fold cross-validation by splitting the training data to 10 sets. Every time, the decoder is trained on 9 sets for different set of parameters and validated on the last set. We find the set of optimum parameters by using random search, as it has shown better performance than grid search [28]. Finally, we test the decoder with optimized parameters on the test set. The performance on all the test sets is averaged to report the overall performance of the models in both single- and multi-day analysis.

We compare our DRNN with other decoders, ranging from linear and historical decoders to nonlinear and modern techniques. The linear and historical decoders with which we compare ours are the Linear Model (LM) [2] and Kalman Filter (KF) [3]. The nonlinear and modern techniques with which we also compare ours include Support Vector Regression (SVR) [29], Gaussian KARMA [6], tree based algorithms (e.g., XGBoost (XGB) [25, 26, 30], Random Forest (RF) [31], and Decision Tree (DT) [32]), and neural network based algorithms (e.g., Deep Neural Networks (NN) [5], Recurrent Neural networks with simple recurrent units (RNN) [33], Long-Short Term Memory units (LSTM) [34], Gated Recurrent Units (GRU) [35], and F-DRNN [7]). (See supplementary material).

We first present single-day performance of DRNN, which is a common practice in the field [7, 3, 36] and is applicable when the training data is limited to a single day. Moreover, there are aspects that differ between single- and multi-day decoding, which have not yet been well characterized (e.g., varying sources of signal instability) and remain challenging in neuroscience. Furthermore, single-day decoding is important before considering multi-day decoding since our implantable hardware will be developed such that the decoder parameters can be updated at any time.

## 4.1 Single-day performance

We select the MWT as the input neural feature. The models are trained on the first $90\%$ of a day and tested on the remaining $10\%$. Figure 2 shows the average performance of the decoders. History-Less DRNN (HL-DRNN) uses the neural data at time k and kinematics at time k-1 to make predictions at time k. As we see, DRNN and HL-DRNN are more stable and have higher average performance.

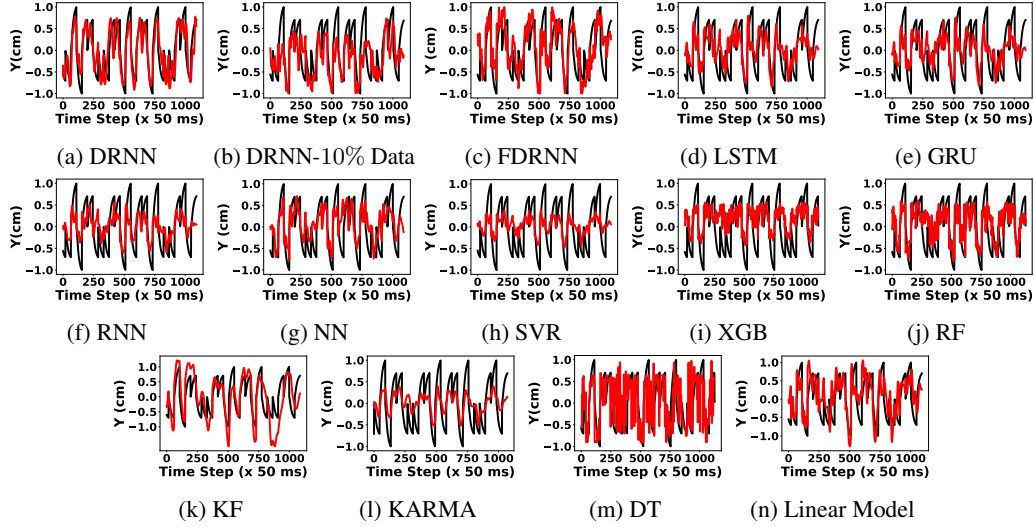

(a) DRNN     (b) DRNN-10% Data     (c) FDRNN     (d) LSTM     (e) GRU

(f) RNN     (g) NN     (h) SVR     (i) XGB     (j) RF

(k) KF     (l) KARMA     (m) DT     (n) Linear Model

Figure 4: Regression of different algorithms on test data from the same day 2018-04-23: true target motion (black) and reconstruction (red).

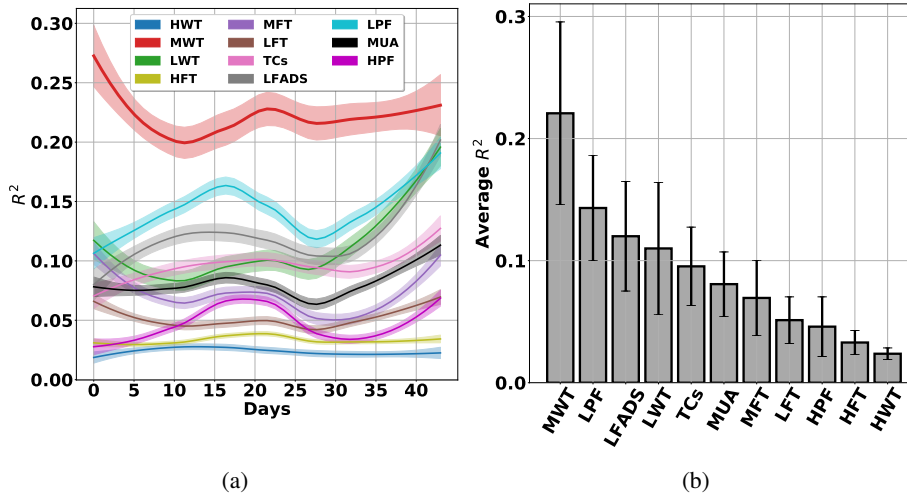

(a)                    (b)

Figure 5: Cross-day analysis of the DRNN.

Figure 4 shows the regression of all the decoders on a sample day. We use only 10% of the single-day training data in figure 4 (b) to show the stability of the DRNN to the limited amount of single-day training data. Other single-day analyses, including evaluation of the DRNN by changing the amount of single-day training data, the history of neural data, and the number of nodes are presented in supplementary material.

For cross-day analysis, we train the DRNN on a single day and test it on all the other days, and repeat this scenario for all the days. Figure 5 shows the performance of the DRNN over all the days. This figure shows that MWT is a more robust feature across single days.

## 4.2 Multi-day performance

To evaluate the effect of the selected feature on the stability and performance of the DRNN, we train the DRNN on the data from the first 20 days of 2015 and test it on the consecutive days by using different features. Figure 6 shows that the DRNN operating on the MWT results in superior performance compared to the other features. Black vertical lines show the year change. We show that the MWT are also the best for a range of decoders in supplementary material.

Then, we evaluate the stability and performance of all the decoders over time. Figure 7 shows that the

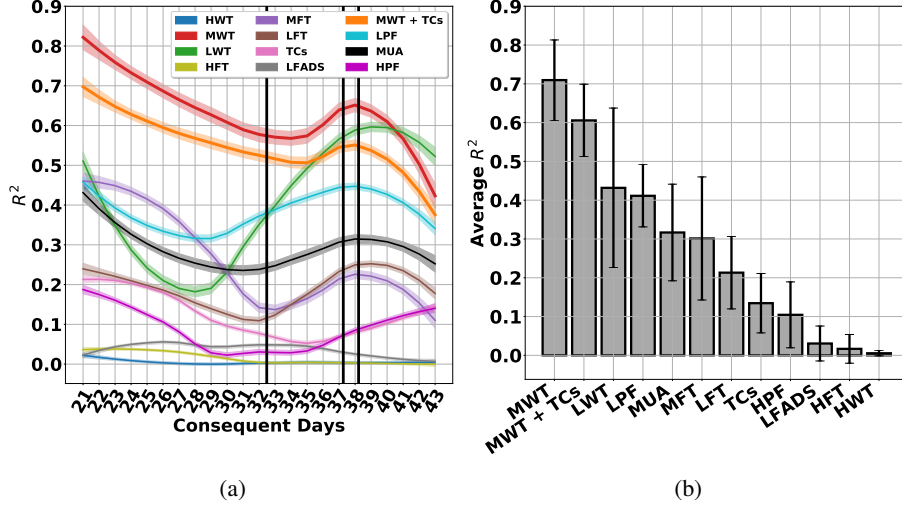

| | | |
|---|---|---|
| (a) | | (b) |

Figure 6: The DRNN operating on different features.

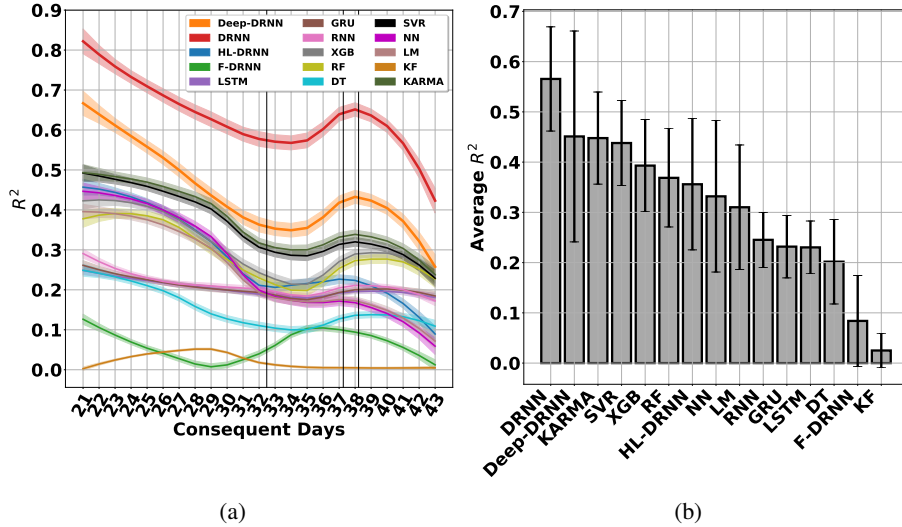

| | | |
|---|---|---|
| (a) | | (b) |

Figure 7: Multi-day performance of the decoders.

overall and the average performance of the DRNN exceeds other decoders. Moreover, the DRNN shows almost stable performance across 3 years. The drop in the performance of almost all the decoders is because of the future neural signal variations [13].

To assess the sensitivity of the decoders to the number of training days, we change the number of training days from 1 to 20 by starting from day 20. Figure 8 shows that the Deep-DRNN with 2 layers and the DRNN have higher performance compared to the other decoders, even by using a small number of training days. Moreover, figure 8 shows that the performance of the DRNN with 1 layer, 10 nodes, and history of 10 is comparable to the Deep-DRNN with 2 layers, 50 and 25 nodes in the first and second layers, and history of 20. Therefore, a small DRNN with a short history has superior performance compared to the other decoders.

To evaluate the effect of re-training the DRNN, we consider four scenarios. First, we train DRNN on the first 20 days of 2015 and test it on the subsequent days. Second, we re-train a DRNN, which has been trained on 20 days, with the first 5%, 10%, 50%, and 90% of the subsequent test days. Third, we re-train the trained DRNN annually with 5%, 10%, 50%, and 90% of the first days of 2016, 2017, and 2018. Finally, we train DRNN only on the first 5% and 90% of the single test day. Figure 9 shows a general increase in the performance of the DRNN after the network is re-trained. The differences between the performances of the first three scenarios are small, which means that the DRNN does not necessarily need to be re-trained to perform well over multiple days. However, because of inherent

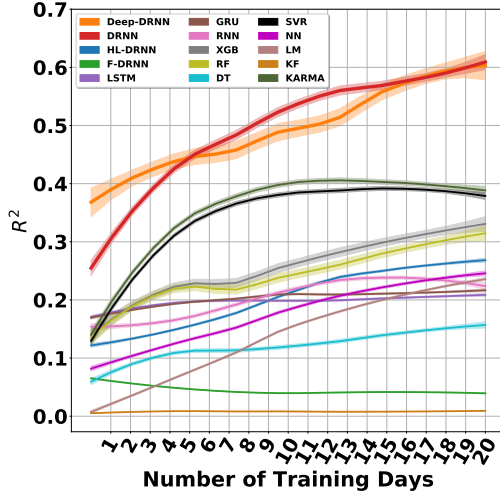

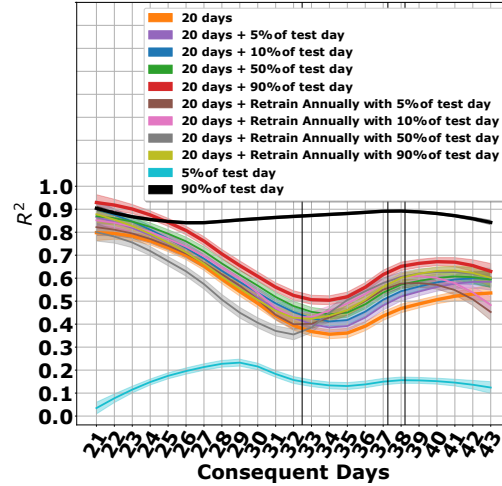

Figure 8: Effect of number of training days on the performance of the decoders.

Figure 9: The DRNN operating in different training scenarios.

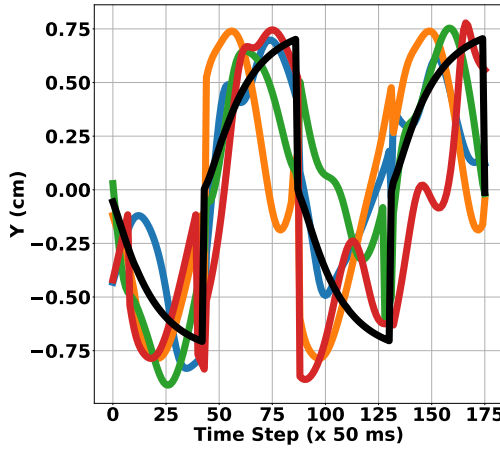

(a)

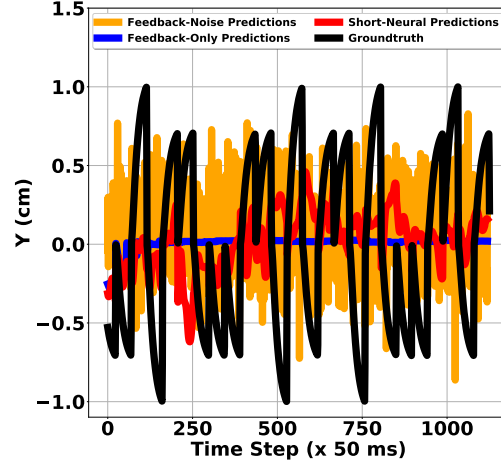

(b)

Figure 10: (a) DRNN predictions for sample targets in all four quadrants, (b) DRNN predictions - no/short neural data. True target motion (black) and reconstructions (colored)

nonstationarity of the recorded neural data over multiple days [13], training the DRNN on the first 90% of the same test day in the last scenario results in the highest average test performance.

The DRNN relies on neural data inputs–not just the kinematic feedback or target information–based on the following evidence. First, target information is not explicitly provided to the DRNN. Any target information available to the DRNN is learned from the neural data and/or feedback components. Second, DRNN outputs change substantially based on different feature engineering approaches (Figures 5, 6) and over different trials (with the same features) (Figures 4, 10a). Finally, predictions fail when the DRNN uses only feedback (Feedback-Only), feedback with noise substituted for neural data (Feedback-Noise), or feedback with the neural data provided only at the beginning of the trials (Short-Neural) (Figure 10b).

## 5 Hardware implementation potential

BMIs are intended to operate as wireless, implantable systems that require low-power circuits, small physical size, wireless power delivery, and low temperature deltas ($\leq 1°C$) [37, 26, 38]. By choosing efficient algorithms that map well to CMOS technologies, Application Specific Integrated Circuit (ASIC) implementations could offer substantial power and mobility benefits. We are proposing

a method that will not only have good performance on single- and multi-day data, but will also be optimal for hardware implementation. Since it is impractical to require powerful CPUs and GPUs for everyday usage of a BMI device, we need a device that is easily portable and does not require communication of the complete signals recorded by electrodes to an external computer for computation. Doing the computation in an ASIC would drastically reduce the latency of kinematics inference and eliminate a large power draw for the gigabytes of neural data that must be transferred otherwise. Thus, we plan to create an ASIC that can be implanted in the brain to perform inference of kinematics from neural signals. The main bottleneck in most neural network accelerators is the resources spent on fetching input history and weights from memory to the Multiplication and Accumulation (MAC) unit [39]. The DRNN will help mitigate this issue since it requires fewer nodes and input history compared to the standard recurrent neural networks. This eliminates the need for large input history storage and retrieval, reducing latency and control logic. Furthermore, by using 16-bit fixed point values for the weights and inputs rather than floating point values, we can reduce the power used by the off-chip memory [39, 40].

## 6    Discussion

We propose a Deep Multi-State DRNN with feedback and scheduled sampling to better model the nonlinearity between the neural data and kinematics in BMI applications. We show that feeding back the DRNN output recurrently result in better performance/more robust decodes. Feeding the output back to the input recurrently in addition to the input neural data provides more information to the DRNN to make predictions, which results in a smaller network with less history. Analogous to the gain term of the Kalman filter, the DRNN learns the relative importance of the neural data and feedback. Integrating both state and neural information in this way leads to smoother predictions (Figure 4a). In addition, we show that the added internal derivative state enables our DRNN to track first order and more complicated patterns in the data. Our DRNN is unique since it learns a matrix that establishes a relationship between the past and present derivative states unlike the conventional DRNN. Also our DRNN, which learns all the model parameters by using back propagation through time (BPTT), is distinct from F-DRNN as the most similar previous model in BMI, which only learns the output weight by using RLS algorithm. Moreover, its application differs from most of the existing decoders that have been applied to motor cortex data of a non-human primate. To the best of our knowledge, we present the first demonstration of applying feedback and scheduled sampling to a DRNN and comparing different learning based decoders operating on different features to predict kinematics by using open-loop neural data recorded from the PPC area of a human subject in a real BMI setting. Our DRNN has the potential to be applied to the recorded data from other brain areas as a recurrent network.

To evaluate our DRNN, we analyze single-day, cross-day, and multi-day behavior of the DRNN by extracting 12 different features. Moreover, we compare the performance and robustness of the DRNN with other linear and nonlinear decoders over 43 days. Results indicate that our proposed DRNN, as a nonlinear dynamical model operating on the MWT, is a powerful candidate for a robust BMI.

The focus of this work is to first evaluate different decoders by using open-loop data since the data presented was recorded from a subject who has completed participation in the clinical trial and has had the electrodes explanted. However, the principles learned from this analysis will be relevant to the future subjects with electrodes in the same cortical area.

Future studies will evaluate the DRNN performance in a closed-loop BMI, in which all the decoders use the brain's feedback. Next, since we believe that our small DRNN achieves higher efficiency and uses less memory by reducing the history of the input, number of weights, and therefore memory accesses, we are planning to implement the DRNN in a field-programmable gate array (FPGA) system where we can optimize for speed, area, and power usage. Then, we will build an ASIC of the DRNN for BMI applications. The system implemented must be optimized for real-time processing. The hardware will involve designing multiply-accumulates with localized memory to reduce the power consumption associated with memory fetch and memory store.

**Acknowledgment:**    We thank Tianqiao and Chrissy (T&C) Chen Institute for Neuroscience at California Institute of Technology (Caltech) for supporting this IRB approved research. We also thank Dr. Erin Burkett for reviewing this manuscript.

## Footnotes

[1]Our code is available at: https://github.com/BenyaminHaghi/DRNN-NeurIPS2019

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
