[Supplementary Material]

# Supplementary Material: Deep Multi-State Dynamic Recurrent Neural Networks Operating on Wavelet Based Neural Features for Robust Brain Machine Interfaces

**Benyamin Haghi**[1,*], **Spencer Kellis**[2], **Sahil Shah**[1], **Maitreyi Ashok**[1], **Luke Bashford**[2], **Daniel Kramer**[3], **Brian Lee**[3], **Charles Liu**[3], **Richard A. Andersen**[2], **Azita Emami**[1]

1 Electrical Engineering Department, Caltech, Pasadena, CA, USA
2 Biology and Biological Engineering Department, Caltech, Pasadena, CA, USA
3 Neurorestoration Center and Neurosurgery, USC Keck School of Medicine, L.A., CA, USA

*benyamin.a.haghi@caltech.edu

## 1 Dynamic recurrent neural network

A general structure of a discrete-time DRNN is given by the following expressions:

$$\begin{cases} s_k = -as_{k-1} + f(W_{ss}, s_{k-1}, W_{su}, u_k, b_s) \\ \hat{y}_k = W_{ys}s_k + b_y \end{cases} \tag{1}$$

where $s \in \mathbb{R}^N$, $\hat{y} \in \mathbb{R}^M$, and $u \in \mathbb{R}^I$ are the state, prediction, and the input vectors, respectively, $W_{ss} \in \mathbb{R}^{N \times N}$, $W_{su} \in \mathbb{R}^{N \times I}$, and $W_{ys} \in \mathbb{R}^{M \times N}$ are the weight matrices, $a \in [-1, 1]$ is a constant controlling state decaying, $b_s \in \mathbb{R}^N$, and $b_y \in \mathbb{R}^M$ are the biases, and $f : \mathbb{R}^N \times \mathbb{R}^I \to \mathbb{R}^N$ is a vector-valued function. [1]

## 2 Approximation of state-space trajectories

Theorem 2.1 verifies the approximation capability of DRNNs for the discrete-time, and non-linear systems.

**Theorem 2.1** *Let $S \subset \mathbb{R}^M$ and $U \subset \mathbb{R}^I$ be open sets, $D_s \subset S$ and $D_u \subset U$ be compact sets, and $f : S \times U \to \mathbb{R}^M$ be a continuous vector-valued function which defines the following non-linear system*

$$z_k = f(z_{k-1}, u_k), z \in \mathbb{R}^M, u \in \mathbb{R}^I \tag{2}$$

*with an initial value $z_0 \in D_s$. Then for an arbitrary number $\epsilon > 0$, and an integer $0 < L < \infty$, there exist an integer N and a DRNN of the form (1) with an appropriate initial state $s_0$ such that for any bounded input $u : \mathbb{R}^+ = [0, +\infty) \to D_u$*

$$\max_{0 \le k \le L} ||z_k - s_k|| < \epsilon \tag{3}$$

Proof: See [1]

## 3 Local stability and convergence of DRNNs

Learning rate ($\gamma$) plays the main role in stability and convergence of neural networks. By using Lyapunov theorem, we define the range of the learning rate to guarantee the real-time convergence of DRNNs and the stability of the system during the whole control process.

**Theorem 3.1** *If an input series of internal dynamic neural network can be activated in the whole control process subject to $u_k \in \mathbb{R}^I$, then learning rate satisfies*

$$0 < \gamma < \frac{2}{r^2} \tag{4}$$

*where $r = \frac{\partial e}{\partial W}$, $e = \hat{y} - y$ is the difference of prediction and ground-truth, and W is the concatenation of connection weights of each network unit. Then (4) ensures the system is exponentially convergent.*

Proof: See [2]

## 4 Description of the other methods

Since all of these methods are well-known in the literature, we only provide a brief explanation of each here. We explain the F-DRNN with details since our network is a generalization of the F-DRNN, with all the parameters to be learnable. For more information, please take a look at the main references. We use Pytorch, Keras, Scikit-learn and Python 2.7 for simulations. [3, 4, 5].

### 4.1 Latent Factor Analysis via Dynamical Systems (LFADS) [6]

Latent Factor Analysis via Dynamical Systems (LFADS) works by modeling a dynamical system that can generate neural data. The algorithm models the nonlinear vector valued function F that can infer firing rates using neural data input. The LFADS system is a generalization of variational auto-encoders that can be used with sequences of data, to model the time-varying aspect of neural signals. We use observed spikes as the input to the encoder RNN. We bin our spikes in 50 ms bins and then separate each center-out task into a separate trial. We use the inferred firing rates that are the result of applying a nonlinearity and affine transformation on the factors output from the generator RNN. A dimensionality of 64 was chosen for the latent variables that are the controller outputs and the factors.

### 4.2 FORCE Dynamic Recurrent Neural Network (F-DRNN) [7]

F-DRNN is defined as below:

$$\begin{cases} \tau \frac{ds_t}{dt} = -s_{t-1} + gW_{sr}r_{t-1} + \beta W_{si}u_t + W_{sf}\hat{y}_{t-1} + b_s \\ r_t = tanh(s_t) \\ \hat{y}_t = W_{yr}r_t + b_y \end{cases} \tag{5}$$

$s \in \mathbb{R}^N$ is the activation variable, and $r \in \mathbb{R}^N$ is the vector of corresponding firing rates. These states track the first and zero order differential features of the system, respectively.

$W_{sr} \in \mathbb{R}^{N \times N}$ describes the relationship between $s$ and $r$. $W_{su} \in \mathbb{R}^{N \times I}$ relates $s$ to the input vector $u$. $\hat{y}$ models the feedback in the network. $W_{sf} \in \mathbb{R}^{N \times M}$ tracks the effect of $\hat{y}$ on $s$. $W_{yr} \in \mathbb{R}^{M \times N}$ indicates the linear transformation between the firing rates $r$ and the prediction $\hat{y}$. $\tau$, $g$, and $\beta$ are the neuronal time constant, scaling of internal connections, and scaling of inputs, respectively.

To discretize the first equation of the continuous F-DRNN, we integrate the first expression of the system by using the Euler method at a time step of $\Delta t$. Then, the equations of the network become:

$$\begin{cases} s_k = (1-c)s_{k-1} + cgW_{sr}r_{k-1} + c\beta W_{si}u_k + cW_{sf}\hat{y}_{k-1} + cb_s \\ r_k = tanh(s_k) \\ \hat{y}_k = W_{yr}r_k + b_y \end{cases} \tag{6}$$

where $c = \frac{\Delta t}{\tau}$

In F-DRNN, we select $W_{sr}$, $W_{si}$, and $W_{sf}$ to be randomly sparse, i.e., only $n = 0.1N$, $i = 0.1I$, and $m = 0.5M$ randomly chosen elements are non-zero in each of their rows, respectively. The non-zero elements of the matrices are drawn independently from zero-mean Gaussian distributions with variances $\frac{1}{n}$, $\frac{1}{i}$, and $\frac{1}{m}$, respectively. $s_0$ and the constant bias $b_s$ are drawn from zero-mean Gaussian distributions with the standard deviations $\sigma_s$ and $\sigma_b$, respectively. Elements of $W_{yr}$ are initialized to zero. Since the only matrix learned in the network is the output weight $W_{yr}$ and all the other

weights are fixed, we name this network as FORCE DRNN [7]. Therefore, the network dynamics are controlled by the matrix $gW_{sr} + W_{sf}W_{yr}$. To update $W_{yr}$, we use recursive least-squares (RLS) algorithm [8]. The error signal is defined by:

$$e_k = \hat{y}_k - y_k \tag{7}$$

By defining $P$ as the inverse correlation matrix, the equation for updating $P$ is as below:

$$P_k = P_{k-\Delta k} - \frac{P_{k-\Delta k}r_k r_k^T P_{k-\Delta k}}{1 + r_k^T P_{k-\Delta k}r_k}, \ P_0 = \gamma I \tag{8}$$

Where $\gamma$ and $\Delta k$ indicate learning rate and learning step size (batch size), respectively. The modification to the $m^{th}$ column of $W_{yr}$ by using the $m^{th}$ element of the error is:

$$W_{yr_k^{(m)}} = W_{yr_{k-\Delta k}}^{(m)} - e_k^{(m)} P_k r_k \tag{9}$$

## 4.3   Deep Neural Network (NN) [9]

In a fully connected neural network, there are multiple layers: an input layer, output layer, and any number of hidden layers with multiple nodes in each hidden layer. The output of each node in each layer is connected to the input of each node in the consecutive layer. Each node performs of $\sum_{i=1}^{N} W_i x_i$, where $x_i$ is each input from the nodes in the previous layer and $W_i$ is the weight of the connection between the node in the previous layer and this current node. The output is then converted to a normalized range using a function such as $tanh$ to get values between -1 and 1. $W_i$ is trained through a process called back-propagation that trains the network on the inputs and finds the error, iteratively minimizing the loss function until the error stays relatively constant.

Since over-fitting is possible, which can cause issues where the trained model cannot later generalize to the separate test data, we can try to perform early stopping during validation such that a limited number of epochs (round of training with all inputs) are used for training before the weights are finalized. The following number of epochs are considered in our work: 5, 10, 20, 30, 50, 75, 100, 125, 150, 200, 300, 400, 500, 600. In addition, we consider different network structures with up to 3 layers, where each set consists of 1, 2, or 3 hidden layers with the given number of nodes in each layer: (100), (100, 100), (100, 10), (20, 20), (20, 20, 20), (40, 40), (40, 10), (40, 40, 40), (10, 10, 10).

## 4.4   Support Vector Regression (SVR) [10]

Support vector regression (SVR) is the continuous form of support vector machines where the generalized error is minimized, given by the function:

$$\hat{y} = \sum_{i=1}^{N} (\alpha_i^* - \alpha_i)k(u_i, u) + b \tag{10}$$

and $\alpha_i$ are Lagrange multipliers and $k$ is a kernel function, where we use the radial basis function kernel in this paper. The Lagrange multipliers are found by minimizing a regularized risk function:

$$\frac{1}{2}||w||^2 + C\sum_{i=1}^{l} L_\epsilon(y) \tag{11}$$

We vary the penalty portion of the error term, C, as part of the validation process to find the optimum parameter.

## 4.5   Linear Model (LM) [11]

The linear model uses a standard linear regression model where we can predict kinematics ($\hat{y}$) from the neural data ($u$) by using:

$$\hat{y} = a + \sum_{i=1}^{N} W_i u_i \tag{12}$$

We find the weights $W_i$ and the bias term $a$ through a least squares error optimization to minimize mean squared error between the model's predictions and true values during training. The parameters are then used to predict new kinematics data given neural data.

### 4.6 Kernel Auto-Regressive Moving Average (KARMA) [12]

The Kernel Auto-Regressive Moving Average (KARMA) model can also be used for prediction. ARMA (non-kernelized) uses the following model, where $\hat{y}_k^i$ is the $i^{th}$ component of the kinematics data at time step $k$ and $u_s^j$ is the $j^{th}$ component of the neural data at time step $s$:

$$\hat{y}_k^i = \sum_{l=1}^{r} A_l \hat{y}_{k-1}^i + \sum_{l=1}^{s} B_l u_{k-l+1}^i + e_k^i \tag{13}$$

Thus, we are performing a weighted average of the past $r$ time steps of kinematics data and the past $s$ time steps of neural data (as well as the current one) with a residual error term, $e$. Then, the difference in KARMA is that we use the kernel method to translate data to the radial basis function dimension. We use a standard SVR solver for inference, by just concatenating the different histories for the kinematics and neural data. When training, we use the known kinematics values for the history. However, when predicting new kimatics data, we use old predictions for the history portion of the new predictions.

### 4.7 Kalman Filter (KF) [13]

The Kalman Filter combines the idea that kinematics are a function of neural firings as well as the idea that neural activity is a function of movements, or the kinematics. This can be represented by two equations:

$$\begin{cases} \hat{y}_{k+1} = A_k \hat{y}_k + w_k \\ u_k = H_k \hat{y}_k + q_k \end{cases} \tag{14}$$

These represent how the system evolves over time as well as how neural activity is generated by system's behavior. The matrices $A$, $H$, $Q$, and $W$ can be found through a training process (where $q \sim \mathcal{N}(0, Q)$ and $w \sim \mathcal{N}(0, W)$). Using properties of the conditional probabilities of kinematics and neural data, we get a closed form solution for maximizing the joint probability $p(Y_M, U_M)$. Using the physical properties of the problem, we get matrix A to be of the form:

$$A = \begin{bmatrix} 1 & 0 & \Delta t & 0 \\ 0 & 1 & 0 & \Delta t \\ 0 & 0 & a & b \\ 0 & 0 & c & d \end{bmatrix} \tag{15}$$

We define $A_v$ as:

$$A_v = \begin{bmatrix} a & b \\ c & d \end{bmatrix} = V_2 V_1^T (V_1 V_1^T)^{-1} \tag{16}$$

$V_1$ consists of the velocity kinematics points except for the last time step, $V_2$ consists of the velocity kinematics points except for the first time step, and $dt$ is the time step size used, 50 ms in our case.

Furthermore $W$ is a zero matrix with the matrix $W_v = \frac{1}{N-1}(V_2 - AV_1)(V_2 - AV_1)^T$ in the bottom right corner. $H$ and $Q$ are given by:

$$\begin{cases} H = U^T Y (YY^T)^{-1} \\ Q = \frac{1}{N}(U - HY)(U - HY)^{-1} \end{cases} \tag{17}$$

Then, we can use the update equations:

$$\begin{cases} \hat{y}_k^- = A \hat{y}_{k-1} \\ P_k^- = AP_{k-1}A^T + W \\ \hat{y}^k = \hat{y}_k^- + K_k(u_k - H\hat{y}_k^-) \\ P_k = (1 - K_k H)P_k^- \end{cases} \tag{18}$$

Here, P is the covariance matrix of the kinematics. $K_k$, the Kalman filter gain is given by:

$$K_k = P_k^- H^T (HP_k^- H^T + Q)^{-1} \tag{19}$$

## 4.8 Recurrent Neural Network (RNN) [14]

A vanilla recurrent neural network with **N** hidden nodes for regression is defined as:

$$\begin{cases} r_k = tanh(W_{rr}r_{k-1} + W_{ri}u_k + b_r) \\ \hat{y}_k = W_{yr}r_k + b_y \end{cases} \tag{20}$$

where $r \in \mathbb{R}^N$, $\hat{y} \in \mathbb{R}^M$, and $u \in \mathbb{R}^I$ are the state, prediction, and input vectors, respectively, $W_{rr} \in \mathbb{R}^{N \times N}$, $W_{ru} \in \mathbb{R}^{N \times I}$, and $W_{yr} \in \mathbb{R}^{M \times N}$ are the weight matrices, $b_r \in \mathbb{R}^N$ and $b_y \in \mathbb{R}^M$ are the biases.

Because of the internal state $r$ which acts as a history unit, the RNN is capable of remembering and extracting short term temporal dependencies in sequential data. Therefore, to find the spatio-temporal relationship between the recorded neural data and kinematics as sequential data, we train an RNN with optimal parameters and compare its performance with the DRNN.

## 4.9 Long-Short Term Memory Recurrent Neural Network (LSTM) [15]

It is well-known that Simple RNN units cannot remember long term dependencies in sequential data because of the vanishing gradients problem. Another version of RNNs that is widely used in the literature are RNNs with Long-Short Term Memory (LSTM) units. By denoting $\circ$ as Hadamard product, the LSTM is defined as:

$$\begin{cases} f_k = \sigma(W_{fu}u_k + W_{fr}r_{k-1} + b_f) \\ i_k = \sigma(W_{iu}u_k + W_{ir}r_{k-1} + b_i) \\ o_k = \sigma(W_{ou}u_k + W_{or}r_{k-1} + b_o) \\ c_u = tanh(W_{cu}u_k + W_{cr}r_{k-1} + b_c) \\ c_k = f_k \circ c_{k-1} + i_k \circ c_u \\ r_k = o_k \circ tanh(c_k) \\ \hat{y}_k = W_{yr}r_k + b_y \end{cases} \tag{21}$$

$r_k$ is the hidden state as in Simple RNN, $c_u$ is the output from the cell update activation function, $c_k$ is the LSTM cell's internal state, $f_k$, $i_k$, $o_k$ are the output matrices from the respective forget, input, and output activation functions, which act as the LSTM's gates, $W$ and $b$ represent the weights and biases, and $\sigma$ is the sigmoid function.

## 4.10 Gated Recurrent Units (GRU) [16]

A simpler version of the RNN cells than LSTM that can extract long term dependencies in sequential data are Gated Recurrent Units (GRU). The GRU formulation is as below:

$$\begin{cases} z_k = \sigma(W_{zu}u_k + W_{zr}r_{k-1} + b_z) \\ h_k = \sigma(W_{hu}u_k + W_{hr}r_{k-1} + b_h) \\ r_u = tanh(W_{ru}u_k + W_{rr}(h_k \circ r_{k-1}) + b_r) \\ r_k = (1 - z_k) \circ r_{k-1} + z_k \circ r_u \\ \hat{y}_k = W_{yr}r_k + b_y \end{cases} \tag{22}$$

Here, $h$ is a reset gate, and $z$ is an update gate. The reset gate determines how to combine the previous memory and the new input. The network decides how much of the previous memory should be kept by using the update gate. Vanilla RNN is the case that we set the update gate to all 0's and the reset to all 1's.

## 4.11 XGBoost (XGB) [17, 18]

XGBoost is one kind of boosting methods which uses ensemble of decision trees. Among 29 competitions winning solutions published at Kaggle during 2015, 17 solutions used XGBoost [17]. For a given data set with $n$ examples and $m$ features $D = \{(x_i, y_i)\}, |D| = n, x_i \in \mathbb{R}^m, y_i \in \mathbb{R}$, a tree ensemble model uses $K$ additive functions to predict the output:

$$\hat{y}_i = \rho(x_i) = \sum_{k=1}^{K} f_k(x_i), f_k \in F \tag{23}$$

where $F = \{f(x) = w_{q(x)}\}, (q : \mathbb{R}^m \to T, w \in \mathbb{R}^T)$ is the space of regression trees, $q$ represents the structure of each tree, $T$ is the number of leaves, and each $f_k$ corresponds to a tree structure q and leaf weights w.

## 4.12  Random Forests (RF) and Decision Trees (DT) [19, 20]

Random Forests are one kind of bagging tree based algorithms that make the prediction by routing a feature sample through the tree to the leaf randomly. The training process will be done independently for each tree. The forest final prediction is the average of the predictions of all the trees. Decision trees are a special case of random forests with one tree.

## 5  DRNN training: back propagation through time (BPTT)

The squared loss function is defined as below:

$$J = \frac{1}{2}(\hat{y} - y)^2 \tag{24}$$

Therfore, the average loss at time step $k$ is:

$$J_k = \frac{1}{k} \sum_{t=1}^{k} J_t \tag{25}$$

Training the network is usually accomplished by applying a mini-batch optimization method to search for a set of parameters that maximize the log-likelihood function:

$$\theta^* = \arg\max_{\theta} \sum_{(x^i, y^i)} log\, P(y^i | x^i; \theta) \tag{26}$$

$(x^i, y^i)$ is a training pair, $P$ is the probability distribution of the data, and $\theta^*$ is the optimum set of parameters.

To update the model's parameters, we perform back propagation through time (BPTT) by using Adam optimization algorithm [21] (Supplementary materials). We need to find the following derivatives:

$$\frac{\partial J_k}{\partial W_{ss}}, \frac{\partial J_k}{\partial W_{sr}}, \frac{\partial J_k}{\partial W_{si}}, \frac{\partial J_k}{\partial W_{sf}}, \frac{\partial J_k}{\partial b_s}, \frac{\partial J_k}{\partial W_{yr}}, \frac{\partial J_k}{\partial b_y}$$

Since the partial derivative is a linear operator, we have:

$$\frac{\partial J_k}{\partial \theta} = \frac{1}{k} \sum_{t=1}^{k} \frac{\partial J_t}{\partial \theta} \tag{27}$$

where $\theta$ indicates weights or biases.

$\frac{\partial J_t}{\partial W_{yr}}$ and $\frac{\partial J_k}{\partial b_y}$ depend only on the variables at the present time. Therefore, by using the chain rule, we get

$$\frac{\partial J_t}{\partial \theta} = \frac{\partial J_k}{\partial \hat{y}_k} \frac{\partial \hat{y}_k}{\partial \theta} \tag{28}$$

where $\theta \in \{W_{yr}, b_y\}$

The derivatives of $J$ with respect to all the other parameters depend not only on the present time variables, but also they depend on the history of the variables from the beginning of the update time (batch size). Therefore, we have:

$$\frac{\partial J_t}{\partial \theta} = \sum_{i=0}^{t} \frac{\partial J_t}{\partial r_t} \frac{\partial r_t}{\partial r_i} \frac{\partial r_i}{\partial s_i} \frac{\partial s_i}{\partial \theta} \tag{29}$$

where $\theta \in \{W_{ss}, W_{sr}, W_{si}, W_{sf}, b_s\}$

BPTT has been implemented by using Pytorch automatic differentiation package [5].

# 6  Adam optimization method [21]

Adam optimization algorithm is widely used for parameter optimization in neural network based learning methods. The algorithm is as below:

---
**Algorithm 1** Adam Stochastic Optimization Algorithm

---
Require: $\gamma$: Learning rate
Require: $\beta_1$, $\beta_2 \in [0,1)$: Exponential decay rates for moment estimates
Require: $J(\theta)$: Loss function with parameter $\theta$
Require $\theta_0$: Initial parameter vector, $m_0 \leftarrow 0$ (Initialize $1^{st}$ moment vector), $v_0 \leftarrow 0$ (Initialize $2^{nd}$ moment vector), $t \leftarrow 0$ (Initialize timestep)
**while** $\theta_t$ not converged **do**
    $t \leftarrow t + 1$
    $g_t \leftarrow \nabla_\theta J_t(\theta_{t-1})$ (Get gradients w.r.t. stochastic objective at timestep t)
    $m_t \leftarrow \beta_1 m_{t-1} + (1 - \beta_1)g_t$ (Update biased $1^{st}$ moment estimate)
    $v_t \leftarrow \beta_2 v_{t-1} + (1 - \beta_2)g_t^2$ (Update biased $2^{nd}$ raw moment estimate)
    $\hat{m}_t \leftarrow \frac{m_t}{(1-\beta_1^t)}$ (Compute bias-corrected $1^{st}$ moment estimate)
    $\hat{v}_t \leftarrow \frac{v_t}{(1-\beta_2^t)}$ (Compute bias-corrected $2^{nd}$ raw moment estimate)
    $\theta_t \leftarrow \theta_{t-1} - \gamma \frac{\hat{m}_t}{(\sqrt{\hat{v}_t}+\epsilon)}$ (Update parameters)
**end while**
return $\theta_t$ (Resulting parameters)

---

# 7  Performance evaluation measures

As a pre-processing step before passing the neural data to the decoders, we use XGBoost feature importance score to select stable channels across the training days. The more a feature is used to make key decisions with XGBoost decision trees, the higher its relative importance. This importance is calculated explicitly for each feature in the dataset, allowing features to be ranked and compared to each other. Importance is calculated for a single decision tree by the amount that each feature split point improves the performance measure, weighted by the number of observations the node is responsible for. The importances are then averaged across all of the the XGBoost decision trees.

To compare the decoders' predictions, we report Root Mean Square Error (RMSE), and $R^2$. The RMSE finds the average point-wise error between the predicted and ground-truth signals as below:

$$RMSE = \sqrt{\frac{1}{K}\sum_{i=1}^{K}(y_i - \hat{y}_i)^2} \tag{30}$$

where $K$ is the total number of data points, $y_i$ and $\hat{y}_i$ are the $i^{th}$ ground-truth and prediction, respectively. The smaller the RMSE is, the better the performance.

The $R^2$ measures the strength of the linear association between the predicted and the ground-truth signals as below:

$$R^2 = \left(\frac{\sum_i(y_i - \bar{y})(\hat{y}_i - \bar{\hat{y}})}{\sqrt{\sum_i(y_i - \bar{y})^2}\sqrt{\sum_i(\hat{y}_i - \bar{\hat{y}})^2}}\right)^2 \tag{31}$$

that is a real number varies from 0 to 1. The larger the $R^2$ is, the better the performance.

# 8  Experimental results for other kinematics

Table 1 shows the parameters of all the decoders for single- and multi-day analysis. The average performance of a small DRNN with a short history surpasses the other decoders' in terms of $R^2$ and RMSE. Even the HL-DRNN's results are comparable with the other larger recurrent models with longer histories. Figures 1, 2, and 3 show the single-day predictions of all the decoders on a

sample day for Vy, X, and Vx, respectively. Figures 4 and 5 show the average performance of all the decoders.

Table 1: Optimum parameters for different algorithms (Only differences are reported for multi-day)

| MODEL | SINGLE-DAY | MULTI-DAY |
|---|---|---|
| SVR | $C$ : 0.1, Kernel: RBF | $C$ : 1 |
| KARMA | $r$ : 0, $s$ : 20, $C$ : 0.1, Kernel: Gaussian | $r$ : 0, $s$ : 2, $C$ : 0.1 |
| XGB | number of trees: 15, maximum depth: 8 | number of trees: 20 |
| RF | number of trees: 20, maximum depth: 10 | number of trees: 40 |
| DT | maximum depth: 10 | - |
| NN | Layer: 2, Optimizer: Adam, Nodes: (40, 10), Batch size: 64, dropout: 0, epoch: 118 | Batch size: 128 dropout: 0.25 |
| RNN | Optimizer: RMSprop, Nodes: 25, Batch size: 64 History: 20, dropout: 0.2, epoch: 19 | Nodes: 100, Batch size: 128 History: 40, epoch: 50 |
| LSTM | Optimizer: RMSprop, Nodes: 50, Batch size: 64 History: 40, dropout: 0.35, epoch: 17 | Nodels 75, Batch size: 128 epoch: 50 |
| GRU | Optimizer: RMSprop, Nodes: 75, Batch size: 32 History: 40, dropout: 0.3, epoch: 18 | Batch size: 64 |
| FDRNN | $g$ : 1, $\beta$ : 0.5, Nodes: 1200, Batch size: 10 $\sigma_b$: 0.025, $\sigma_s$: 0.01, $\tau$: 250 ms, epoch: 10 | $g$: 0.5, Nodes: 1500 |
| DRNN | Layer: 1, Optimizer: Adam, **Nodes: 5**, $p_s$: 0.25, $p_f$: 0 Batch size: 16, **History: 10**, dropout: 0.25, **epoch: 2** | **Nodes: 10**, $p_s$: 0.5 Batch size: 64, **epoch: 5** |

We do three more single-day analyses. First, we evaluate the effect of neural data history on the recurrent networks. Figures 6 and 7 show the performance of recurrent decoders versus history of neural data. The performance of other recurrent decoders drops, however, the DRNN's performance is stable. Second, we evaluate the performance of the recurrent networks with different number of internal hidden nodes. Figures 8 and 9 show the performance of these decoders versus the number of hidden nodes. The DRNN's performance has little changes when we add more nodes to the network. Moreover, it still performs superior to the other recurrent decoders. Third, we assess the stability of the recurrent networks to the amount of the single day's training data. Figures 10 and 11 show that the DRNN's performance is still better than the others. Moreover, it implies that the DRNN is a stable decoder and it works well even with less than $10\%$ single-day training data.

Since we have only presented the $R^2$ values for Y-axis position in the body of the paper, we first present the corresponding RMSE values in figures 12, 13, 14, 15, and 16. These figures evaluate the average cross-day performance of the decoders by operating on different features, average multi-day performance of the DRNN on different features, average multi-day performance of the decoders operating on the MWT, effect of the used number of training days on the average performance of the decoders, and the effect of re-training on the DRNN's long term stability and performance in different scenarios, respectively.

Subsequent figures present the equivalent $R^2$ and RMSE values for all the other kinematics. Figures 17 and 18 show the cross-day performance of the decoders. Figures 19 and 20 show the multi-day performance of the DRNN on the different features. Figures 21 and 22 show the performance of all the decoders operating on the MWT. Figures 15, 23, 25, 27, 29, 31, and 33 show the effect of number of training days on the performance of all the decoders. Finally, figures 16, 24, 26, 28, 30, 32, and 34 show the effect of re-training on the DRNN's long term stability and performance in the different scenarios.

Figure 1: Vy velocity regression of different algorithms on test data from the same day 2018-04-23: true target motion (black) and reconstruction (red).

Figure 2: X position regression of different algorithms on test data from the same day 2018-04-23: true target motion (black) and reconstruction (red).

Figure 3: Vx velocity regression of different algorithms on test data from the same day 2018-04-23: true target motion (black) and reconstruction (red).

(a) Vy

(b) X

(c) Vx

Figure 4: Average performance of the decoders operating on the MWT over single-day data.

(a) Y

(b) Vy

(c) X

(d) Vx

Figure 5: Average RMSE of decoders operating on the MWT over single-day data.

(a) Y

(b) Vy

(c) X

(d) Vx

Figure 6: Performance of the recurrent networks versus the amount of single-day history of neural data.

(a) Y

(b) Vy

(c) X

(d) Vx

Figure 7: RMSE of the recurrent networks versus the amount of single-day history of neural data.

(a) Y

(b) Vy

(c) X

(d) Vx

Figure 8: Performance of the recurrent networks on single-day data versus number of internal hidden nodes of the networks.

Figure 9: RMSE of the recurrent networks on single-day data versus number of internal hidden nodes of the networks.

(a) Y

(b) Vy

(c) X

(d) Vx

(a) Y

(b) Vy

(c) X

(d) Vx

Figure 10: The effect of the amount of single-day training data on the performance of the recurrent networks.

Figure 11: The effect of the amount of single-day training data on the performance of the recurrent networks.

(a) Y

(b) Y

Figure 12: Cross-day analysis of the DRNN.

(a) Y

(b) Y

Figure 13: The DRNN operating on different features.

(a) Y

(b) Y

Figure 14: Multi-day performance of the decoders.

Figure 15: Effect of number of training days on the performance of decoders - Y.

Figure 16: The DRNN operating in different training scenarios - Y.

(a) Vy

(b) Vy

(c) X

(d) X

(e) Vx

(f) Vx

Figure 17: Cross-day analysis of the DRNN.

(a) Vy

(b) Vy

(c) X

(d) X

(e) Vx

(f) Vx

Figure 18: Cross-day analysis of the DRNN.

(a) Vy

(b) Vy

(c) X

(d) X

(e) Vx

(f) Vx

Figure 19: The DRNN operating on different features.

(a) Vy

(b) Vy

(c) X

(d) X

(e) Vx

(f) Vx

Figure 20: The DRNN operating on different features.

(a) Vy

(b) Vy

(c) X

(d) X

(e) Vx

(f) Vx

Figure 21: Multi-day performance of the decoders.

(a) Vy

(b) Vy

(c) X

(d) X

(e) Vx

(f) Vx

Figure 22: Multi-day performance of the decoders.

Figure 23: Effect of number of training days on the performance of decoders - Vy.

Figure 24: The DRNN operating in different training scenarios - Vy.

Figure 25: Effect of number of training days on the performance of decoders - Vy.

Figure 26: The DRNN operating in different training scenarios - Vy.

Figure 27: Effect of number of training days on the performance of decoders - X.

Figure 28: The DRNN operating in different training scenarios - X.

Figure 29: Effect of number of training days on the performance of decoders - X.

Figure 30: The DRNN operating in different training scenarios - X.

Figure 31: Effect of number of training days on the performance of decoders - Vx.

Figure 32: The DRNN operating in different training scenarios - Vx.

Figure 33: Effect of number of training days on the performance of decoders - Vx.

Figure 34: The DRNN operating in different training scenarios - Vx.

(a)                                        (b)

Figure 35: (a) DRNN predictions for sample targets in all four quadrants, (b) DRNN predictions - no/short neural data. True target motion (black) and reconstructions (colored) - Vy

(a)                                        (b)

Figure 36: (a) DRNN predictions for sample targets in all four quadrants, (b) DRNN predictions - no/short neural data. True target motion (black) and reconstructions (colored) - X

(a)                                        (b)

Figure 37: (a) DRNN predictions for sample targets in all four quadrants, (b) DRNN predictions - no/short neural data. True target motion (black) and reconstructions (colored) - Vx

# 9 Performance of all the decoders operating on different features

## 9.1 Long-Short Term Memory Recurrent Neural Network (LSTM)

(a) Y

(b) Y

(c) Vy

(d) Vy

(e) X

(f) X

(g) Vx

(h) Vx

Figure 38: The LSTM operating on different features

(a) Y

(b) Y

(c) Vy

(d) Vy

(e) X

(f) X

(g) Vx

(h) Vx

Figure 39: The LSTM operating on different features

## 9.2 Gated Recurrent Units (GRU)

(a) Y

(b) Y

(c) Vy

(d) Vy

(e) X

(f) X

(g) Vx

(h) Vx

Figure 40: The GRU operating on different features

(a) Y

(b) Y

(c) Vy

(d) Vy

(e) X

(f) X

(g) Vx

(h) Vx

Figure 41: The GRU operating on different features

## 9.3 Recurrent Neural Network (RNN)

Figure 42: The RNN operating on different features

(a) Y

(b) Y

(c) Vy

(d) Vy

(e) X

(f) X

(g) Vx

(h) Vx

Figure 43: The RNN operating on different features

## 9.4 FORCE Dynamic Recurrent Neural Network (F-DRNN)

(a) Y

(b) Y

(c) Vy

(d) Vy

(e) X

(f) X

(g) Vx

(h) Vx

Figure 44: The F-DRNN operating on different features

(a) Y

(b) Y

(c) Vy

(d) Vy

(e) X

(f) X

(g) Vx

(h) Vx

Figure 45: The F-DRNN operating on different features

## 9.5 Deep Neural Network (NN)

Figure 46: The NN operating on different features

(a) Y

(b) Y

(c) Vy

(d) Vy

(e) X

(f) X

(g) Vx

(h) Vx

Figure 47: The NN operating on different features

## 9.6 Support Vector Regression (SVR)

(a) Y        (b) Y

(c) Vy        (d) Vy

(e) X        (f) X

(g) Vx        (h) Vx

Figure 48: The SVR operating on different features

Figure 49: The SVR operating on different features

## 9.7 XGBoost (XGB)

Figure 50: The XGB operating on different features

(a) Y

(b) Y

(c) Vy

(d) Vy

(e) X

(f) X

(g) Vx

(h) Vx

Figure 51: The XGB operating on different features

## 9.8 Random Forests (RF)

(a) Y      (b) Y

(c) Vy      (d) Vy

(e) X      (f) X

(g) Vx      (h) Vx

Figure 52: The RF operating on different features

(a) Y

(b) Y

(c) Vy

(d) Vy

(e) X

(f) X

(g) Vx

(h) Vx

Figure 53: The RF operating on different features

## 9.9 Decision Tree (DT)

(a) Y

(b) Y

(c) Vy

(d) Vy

(e) X

(f) X

(g) Vx

(h) Vx

Figure 54: The DT operating on different features

(a) Y

(b) Y

(c) Vy

(d) Vy

(e) X

(f) X

(g) Vx

(h) Vx

Figure 55: The DT operating on different features

## 9.10 Kernel Auto-Regressive Moving Average (KARMA)

(a) Y

(b) Y

(c) Vy

(d) Vy

(e) X

(f) X

(g) Vx

(h) Vx

Figure 56: The KARMA operating on different features

(a) Y

(b) Y

(c) Vy

(d) Vy

(e) X

(f) X

(g) Vx

(h) Vx

Figure 57: The KARMA operating on different features

## 9.11 Kalman Filter (KF)

(a) Y

(b) Y

(c) Vy

(d) Vy

(e) X

(f) X

(g) Vx

(h) Vx

Figure 58: The KF operating on different features

(a) Y

(b) Y

(c) Vy

(d) Vy

(e) X

(f) X

(g) Vx

(h) Vx

Figure 59: The KF operating on different features

## 9.12 Linear Model (LM)

Figure 60: The LM operating on different features

(a) Y         (b) Y

(c) Vy         (d) Vy

(e) X         (f) X

(g) Vx         (h) Vx

Figure 61: The LM operating on different features