[Reviews · NeurIPS 2019]

Reviewer 1



In this paper, the authors present a multi-state Dynamic Recurrent Neural Network architecture and training framework for Brain Machine Interface (BMI), including incorporating scheduled sampling and testing diverse neural features as input. The authors robustly analyze this model in comparison to other prior modeling frameworks on human posterior parietal cortical activity (PPC). This paper is of an impressive quality, containing rigorous and methodical analyses showing clear and significant improvements of their model. The authors compare to twelve baseline models and investigate many aspects of the modeling framework, including single-day vs multi-day performance, generalization of single-day training to other days, the reliance on amount of training data, the optimal preprocessing of neural feature inputs, and generalization of the models over time with different styles of retraining. The paper was very well-written, with most choices and details clearly explained. I also appreciated that the authors showed examples of the regression performance (Figure 4) instead of just reporting summary statistics. However, I do not think it is stated explicitly enough exactly which parts of the model are novel vs novel in the BMI setting vs pulling from prior BMI literature. The paper could benefit from an expansion of the background section of the Introduction and a citation to a “conventional DRNN” method. The authors did include a description of all baseline models in the Supplementary Material but I still think a more concise description of differences from the most similar previous model (presumably F-DRNN) in the main text would benefit readers. Also, what makes this method multi state? Additionally, this paper applies these methods to a novel brain region, posterior parietal cortex. Other than briefly stating this, the authors do not further discuss PPC or include any analyses or citations about how the decoding performance differed from that of more traditional BMI brain regions such as motor cortex data, which causes this contribution to be much less significant. Some questions: Does watching the cursor move for 3 minutes constitute a trial? Why is there a disconnect between figure 7 and the rightmost point of figure 8? The DRNN and Deep-DRNN have significant differences in figure 7 but in figure 8, when trained similarly on 20 days, they perform similarly. Is there an intuitive explanation as to why the more complex model is doing worse (Deep-DRNN vs DRNN)? It doesn’t seem a clear case of overfitting based on the training days plot. EDIT: I'd like to thank the authors for their thorough rebuttal. I am now even more confident of my high score.

Reviewer 2



I am somewhat unclear how real the hardware problem is. I mean if we can do realtime neural networks with a million inputs, shouldnt a few channels be quite easy? I guess at Neurips I would like to see more computational work. It was very nice to see real-world usage though. I like the comparison with a broad range of decoders.

Reviewer 3



The authors consider an important problem, that of decoding neural signals from the brain for control of external devices, such as a computer cursor. The author’s present a deep recurrent neural network for extracting such intent from neural signals and compare it against many other decoders using offline, prerecorded data from the posterior parietal cortex of a human subject during trials in which the subject watched the cursor move from the center of a workspace to individual targets. The authors compare the performance of their decoder to a host of others, training and testing on the same day, and find their decoder, operating on features extracted with wavelets, outperforms the other decoders. A major focus of this work is that of robustness - being able to use a BMI over multiple days without the need for retraining is noteworthy, as this is an important problem that must be addressed for the clinical translation of BMI systems. The authors extract many different types of features from voltage signals recorded from the electrodes of a Utah array and then train their decoder on data from 1 day and test on the remaining days, finding that wavelet based features produce the most robust performance with their decoder. They also train on the first 20 days in their dataset and test on the remaining days, again finding superior performance with their decoder for wavelet based features. They then assess the sensitivity of their decoder and others to the number of days used to initially train it and find uniformly higher performance for their decoding architecture compared to the others. Finally, they examine the performance of their decoder when retrained with various amounts of data from each day, finding performance improves with retraining. An evaluation of the paper now follows. Originality: I believe the main novelty of this paper lies in the problem it is considering. The problem of robust BCI decoding is still just starting to receive attention. The use of wavelet based features for this, to my knowledge is novel and while the DRNN the authors propose does not appear to be highly novel, the application to the problem of robust BCI in PPC, to the best of my knowledge, is. Quality: The authors do a good job of comparing against many different decoders and considering many different types of features which can be extracted from voltage data recorded from a Utah array. The duration of the dataset they test with is also noteworthy. The clarity of the writing and connection to some of the claims of the paper (see next) could be improved. Clarity: The paper is relatively clear. However, additional description of the motivation for the decoder could help in intuition. For example, why does feeding back the decoders output recurrently result in better performance/more robust decodes? Finally, I struggled with understanding one set of results. In section 4.1 (titled ‘Single-day performance’) the paragraph on lines 192-194 seems to describing training on one day and testing on all other days. If so, why is this under this section title? Significance: The authors are considering an important problem. The finding of new features and decoding architectures which lead to more robust decoding is noteworthy. However, my main concern with the present paper is the difficulty of the task used to evaluate performance. In particular, the trials used for evaluation are when the cursor is under computer control. While I understand the motivation for using these trials for an offline analysis (as closed loop effects would make a later offline analysis difficult), I am worried about the difficulty of decoding these types of trials with a recurrent decoder. In particular, for any given target, was the trajectory the cursor took to that target, identical from trial-to-trial? If so, it seems an effective decoding scheme would be to simply classify which target was selected for a trial and then use the recurrent dynamics of the decoder (without regard for neural activity) to move the cursor in a stereotyped way to the target. In this way, the decoder would only need to classify the direction of initial cursor movement from neural activity at the start of a trial. This is important because it is notoriously difficult to predict the closed-loop performance of a decoder from offline results and having an offline dataset to test with which is too simple may exacerbate this problem. While I fully appreciate it is hard to get continuous valued, ground-truth data in study with a human participant, either (1) providing more details in the manuscript about the variability of trajectories to each trial under computer control or (2) in the absence of such variability, showing the decoders effectiveness in an offline decode of hand-control (from a non-human primate experiment) would improve the strength of the results. Update after author response: I thank the authors for their response. My main concern update task difficulty has been at least somewhat addressed and have updated my score to reflect this.

[Author Response · NeurIPS 2019]

1- "Significance:...difficulty of the task": We will add the following to sec.4: The DRNN relies on neural data inputs–not just the kinematic feedback or target information–based on the following evidence. First, target information is not explicitly provided to the DRNN. Any target information available to the DRNN is learned from the neural data and/or feedback components. Second, DRNN outputs change substantially based on different feature engineering approaches (Figures 5,6) and over different trials (with the same features) (Figure 4, Rebuttal Figure 1a). Finally, predictions fail when the DRNN uses only feedback (Feedback-Only), feedback with noise substituted for neural data (Feedback-Noise), or feedback with the neural data provided only at the beginning of the trials (Short-Neural) (Rebuttal Figure 1b).

2- "Clarity...additional description of the motivation": See sec.2 and lines 239-246. Our DRNN is unique since it learns a matrix that establishes a relationship between the past and present derivative states unlike the conventional DRNN. Also our DRNN, which learns all the hyper-parameters of the model by using back propagation through time (BPTT), is distinct from F-DRNN that only learns the output weight by using RLS algorithm. Moreover, its application differs from existing decoders that have been applied to motor cortex data of a non-human primate. We present the first demonstration of applying feedback and scheduled sampling to a DRNN and comparing different learning based decoders operating on different features to predict kinematics from PPC data of a human subject in the BMI setting.

3- "...how real the hardware problem is...": See sec.5. We will add the following: BMIs are intended to operate as wireless, implantable systems that require low-power circuits, small physical size, wireless power delivery, and low temperature deltas ($\leq 1^\circ$C) (Dewhirst 2003). By choosing efficient algorithms that map well to CMOS technologies, ASIC implementations could offer substantial power and mobility benefits.

4- "...show the wavelet features were best for a range of decoders...": We will add results for all the decoders to supplementary material. Wavelet features show superior results for all the decoders (Example: Rebuttal Figure 1c).

5- "Clarity...why does feeding back the decoders' output recurrently result in better performance/more robust decodes?": We will add the following to sec.2: Feeding the output back to the input recurrently in addition to the input neural data provides more information to the DRNN to make predictions, which results in a smaller network with less history. Analogous to the gain term of the Kalman filter, the DRNN learns the relative importance of the neural data and feedback. Integrating both state and neural information in this way leads to smoother predictions (Figure 4a).

6- "...compare with standard approaches": We did compare (Figures 2, 4, 7, 8) our DRNN with 12 standard decoders that are addressed in (Glaser 2017) including Kalman filter, SVR, XGBoost, NN, RNN, GRU, and LSTM.

7- "...how the decoding performance differed from traditional BMI brain regions": While a full discussion is outside the scope of this paper, we will add the following after line 48: PPC processes a rich set of high-level aspects of movement including sensory integration, planning, and execution (Aflalo 2015) and may encode this information differently (Zhang 2017). These characteristics of PPC differentiate it from other brain areas and, while providing a large amount of information to the decoder, also require new paradigms, such as those discussed here, to extract useful information.

8- Question about trial definition: We will add this definition to sec.4: A trial is one trajectory of the cursor from the center of the screen to one of the eight targets on a unit circle (Figure 3).

9- "...disconnect between Figure 7 and ... Figure 8": There is no disconnect since Figure 7 reports the $R^2$ for single days, whereas Figure 8 shows the $R^2$ averaged over 23 test days by excluding 3 worst days.

10- "...why the more complex model is doing worse?": More layers do not always result in superior performance of neural networks (lines 66-71). See supplementary table 1 for optimum parameters of the DRNN.

11- "Clarity...lines 192-194...": Since training and testing are done on single days in cross-day analysis, these lines do belong under the single-day analysis section.

(a) DRNN predictions for sample targets　(b) DRNN predictions - no/short neural data　(c) LSTM predictions

**Rebuttal Figure 1.** (a) and (b) Comment 1: true target motion (black) and reconstructions (colored), (c) comment 4.

[Meta-Review · NeurIPS 2019]

This paper presents a deep recurrent network for decoding neural signals from the brain of a human participant for the control of a computer cursor. All reviewers thought this was an important problem and appreciated the large-scale comparison against other decoders on a pre-recorded dataset. Reviewer 1 thought the paper was of impressive quality and appreciated the experimental rigor and many aspects that were empirically evaluated. They also thought the paper was well written, but asked for more clarification regarding novelty. Reviewer 2 acknowledged the good results, but questioned the nature of the hardware problem. They asked for comparison to standard approaches. Reviewer 3 had similar concerns to Reviewer 1 in terms of architectural novelty, but like the other reviewers thought the application was significant; particularly towards the clinical translation of BMI systems. They noted that performance on offline pre-recorded data had not always generalize to performance with a decoder under closed-loop control. The authors clarified the uniqueness of their architecture in their feedback. They addressed Reviewer 2’s comments about the hardware problem and confirmed that they had compared to many standard approaches. Following some discussion in the reviewing forum, all reviewers are in favor of accepting the paper. I recommend acceptance. Here is some additional feedback for the authors concerning presentation, based on reviewing this paper with the SAC: - The statement "all hyper-parameters are learnable in our DRNN" is referring to parameters, not hyper-parameters. - Use math mode symbols for operations like \tanh and avoid writing text in math mode (e.g. \text{Scheduled sampling}). Use a single letter index for the epoch counter (rather than ep) and single letter or symbol to represent the total number of epochs (rather than epochs). - Figure 1 is nearly unreadable because it is so compressed (as is the text on the x-axis of Figure 2). Also, Figures 5a, 6a, 7a, 8, and 9 might be unreadable to people with color blindness. - There is a mix of teletype font and roman font text in Algorithm 1. Make the font consistent. Some of the teletype seems to refer to variables (e.g. "number of batches") but other teletype is not (e.g. "Update weights and biases").